



# Global transport of stratospheric aerosol produced by Ruang eruption from EarthCARE ATLID, limb-viewing satellites and ground-based lidar observations

Sergey Khaykin[1], Michaël Sicard[2,3], Thierry Leblanc[4], Tetsu Sakai[5], Nickolay Balugin[6], Gwenaël Berthet[7], Stéphane Chevrier[7], Fernando Chouza[4], Artem Feofilov[8], Dominique Gantois[2], Sophie Godin-Beekmann[1], Arezki Haddouche[8], Yoshitaka Jin[9], Isamu Morino[9], Nicolas Kadygrov[1], Thomas Lecas[7], Ben Liley[10], Richard Querel[10], Ghassan Taha[11,12], Vladimir Yushkov[6].

[1] LATMOS - Laboratoire Atmosphère Observations Spatiales, UVSQ, CNRS, Sorbonne University, Guyancourt, France
[2] LACy, Laboratoire de l'Atmosphère et des Cyclones UMR 8105 CNRS, Université de La Réunion, Météo-France, Saint-Denis, Réunion, France
[3] CommSensLab, Department of Signal Theory and Communications, Universitat Politècnica de Catalunya, Barcelona, Spain
[4] Jet Propulsion Laboratory, California Institute of Technology, Wrightwood, CA, USA
[5] Meteorological Research Institute (MRI-JMA), Tsukuba, Japan
[6] Federal State Budgetary Institution Central Aerological Observatory, Dolgoprudnyi, Moscow oblast, Russian Federation
[7] Laboratoire de Physique et Chimie de l'Environnement et de l'Espace, CNRS UMR 7328, Université d'Orléans, Orléans, France
[8] LMD/IPSL, Sorbonne Université, UPMC Univ Paris 06, CNRS, École Polytechnique, Paris, France
[9] National Institute for Environmental Studies (NIES), Tsukuba, Japan
[10] New Zealand Institute for Earth Science Limited, Lauder, New Zealand
[11] Goddard Earth Sciences Technology and Research (GESTAR) II, Morgan State University, Baltimore, MD 21251, USA
[12] NASA Goddard Space Flight Center, Greenbelt, MD 20771, USA

*Correspondence to Sergey Khaykin (sergey.khaykin@latmos.ipsl.fr)*

**Abstract**

The Atmospheric LIDar (ATLID) instrument of the ESA's Earth Cloud, Aerosol and Radiation Explorer (EarthCARE) satellite mission launched in May 2024 provides high-resolution vertical profiling of aerosols and clouds at 355 nm. Fully operational since July 2024, ATLID has been witness to a significant perturbation of stratospheric aerosol budget following the eruptions of Ruang volcano (Indonesia) in late April 2024. Using ATLID together with limb-viewing satellite instruments (OMPS-LP and SAGE III), we quantify the stratospheric aerosol perturbation generated by the Ruang eruption and characterize
the global transport of volcanic aerosols. To evaluate the ATLID performance in the stratosphere, its data are compared with collocated ground-based lidar observations at various locations in both hemispheres and overpass-coordinated balloon flights carrying AZOR backscatter sonde. The intercomparison with suborbital observations suggests excellent performance of ATLID in the stratosphere and proves its capacity to accurately resolve fine structures in the vertical distribution of stratospheric aerosols. Using various satellite observations, we show that Ruang's eruptive sequence in April 2024 produced
eruptive columns reaching 25 km altitude, and resulted in a doubling of the tropical stratospheric aerosol abundance for several months. The eruption timing in austral Fall and its high-altitude reach fostered efficient poleward transport into the southern extratropics during austral Winter 2024. By the time of the austral Fall 2025, the sulphate aerosols from Ruang have spread across the entire Southern hemisphere and were most probably entrained by the 2025 Antarctic polar vortex, potentially enhancing the polar stratospheric cloud occurrence.




## 1. Introduction

Explosive volcanic eruptions can inject large amounts of sulfur dioxide and ash directly into the stratosphere, where $SO_2$ is oxidized to sulfuric acid ($H_2SO_4$) that nucleates to sub-micron sulfate aerosols (Robock, 2000; Kremser et al., 2016). Owing to negligible wet scavenging and slow gravitational settling in the dry, cloud-poor stratosphere, these particles and fine ash -
when present - can persist for months to years, altering the planetary radiation budget by reflecting incoming shortwave radiation while absorbing near-infrared and trapping some outgoing longwave radiation. The radiative perturbation yields a cooler surface and a warmer lower stratosphere (Stenchikov et al., 1998; Ramachandran, 2000; Robock, 2000) and has been linked to a slowdown of the global hydrological cycle (Robock, 2000; Zuo et al., 2022), episodic Eurasian winter warming (Robock, 2000; Stenchikov et al., 2002), El Niño–like sea-surface temperature responses (Zuo et al., 2018; Sun et al., 2019),
and weakened monsoon circulations. These responses underscore a tight coupling between aerosol microphysics, radiation, and large-scale dynamics.

The magnitude and spatial footprint of the climate perturbation depend on (i) the eruptive composition and sulfur burden, (ii) source latitude, (iii) injection altitude relative to the tropopause, and (iv) the contemporaneous dynamical setting (Kremser et al., 2016; Fairlie et al., 2014; Wu et al., 2017). Sulfur-rich plumes favor rapid production of sulphate aerosols and stronger
radiative forcing (Kremser et al., 2016). The presence of ash can modify chemistry and microphysics, alter particle growth pathways, and locally enhance diabatic heating (Robock, 2000; Vernier et al., 2016; Khaykin et al., 2022a). Tropical injections typically achieve the widest reach: air masses entering the lower stratosphere can be transported interhemispherically and to high latitudes by the Brewer–Dobson circulation (BDC), extending aerosol lifetime and radiative influence (Jones et al., 2017). Higher injection heights further prolong residence time through slower sedimentation and longer transport within the BDC.
Synoptic features - such as cyclones, anticyclones, and jet streams - shape early-stage dispersion and can accelerate hemispheric spreading (Fairlie et al., 2014; Wu et al., 2017).

Despite extensive observations, accurately characterizing the eruption-to-aerosol-to-climate chain remains challenging. Reported injection amounts, heights, and sequences can differ across datasets and retrieval algorithms (e.g. Fromm et al., 2014), and ash–sulfate interactions, size-distribution evolution (nucleation, condensation, coagulation), and background
stratospheric variability introduce additional uncertainty. A key diagnostic of the ensuing radiative impact is the stratospheric aerosol optical depth (SAOD), which integrates changes in particle number, size, and composition; however, constraining SAOD and its spatiotemporal evolution requires consistent multi-sensor observations.

In this paper, we bring together the observations by active and passive satellite sensors, ground-based lidars in both hemispheres and balloon-borne measurements to quantify the stratospheric aerosol perturbation generated by the April 2024
Ruang eruption and characterize the subsequent global transport of volcanic aerosol. This work is the first one to exploit the EarthCARE ATLID observations of stratospheric aerosols and assess ATLID's performance in the stratosphere through intercomparison with collocated suborbital measurements.

## 2. Data and methods.

### 2.1 EarthCARE ATLID

The ATmospheric LIDar (ATLID) is an active, high-spectral-resolution backscatter lidar aboard the Earth Cloud, Aerosol and Radiation Explorer (EarthCARE) satellite, a joint mission of the European Space Agency (ESA) and the Japan Aerospace Exploration Agency (JAXA) (Illingworth et al., 2015; Wehr et al., 2023). Operating at an ultraviolet wavelength of 355 nm, ATLID provides global vertical profiles of aerosols and optically thin clouds. The laser transmitter emits pulses with a nominal
output power of ~35 mJ per pulse at a 51 Hz repetition rate. High-spectral-resolution filtering and polarization-sensitive detection enable separation of Mie (particulate) from Rayleigh (molecular) backscatter, while the polarization subsystem measures the Mie signal in both co-polarized and cross-polarized channels to support discrimination of aerosol and cloud particles. The lidar employs a 620 mm aperture telescope to collect backscattered signals from laser pulses emitted at a 3° off-nadir angle to minimize specular reflections from clouds and the ocean surface. ATLID achieves a vertical resolution of ~100
m below 20 km altitude and ~500 m above 20 km, with horizontal sampling of ~280–305 m (two-shot averaging onboard the satellite).

In concept, ATLID is comparable to the Cloud–Aerosol Lidar with Orthogonal Polarization (CALIOP) aboard the Cloud–Aerosol Lidar and Infrared Pathfinder Satellite Observation (CALIPSO) (Winker et al., 2010), however it incorporates notable enhancements: the use of the high-spectral-resolution lidar (HSRL) technique allows direct separation of Rayleigh and
Mie backscatter without relying on molecular calibration regions. Owing to the narrowband spectral filtering and lower intensity of solar radiance at 355 nm, this technique enables a better signal-to-noise ratio during daytime compared to CALIOP. Moreover, ATLID employs solid-state detectors with greater resilience to radiation-induced noise in the South Atlantic Anomaly (SAA) (Feofilov et al., 2023), a region where CALIOP's measurements were degraded due to increased background counts (Noël et al., 2014).





ATLID's standard products include L1B calibrated backscatter profiles and L2A retrievals from the Aerosol–Cloud Products (A-PRO) processor, providing aerosol and cloud feature masks, particle phase classification, extinction coefficients and lidar ratios (Wehr et al., 2023). Here we use L1B product (BA baseline) and include both nighttime and daytime observations to compute scattering ratio (SR) and stratospheric aerosol optical depth (SAOD) for comparison with collocated suborbital measurements and near-global limb-viewing satellite observations. SR is computed as a sum of the three

measurements channels divided by the Rayleigh channel:

$$SR = \frac{Mie_{copolar} + Mie_{crosspolar} + Rayleigh}{Rayleigh}. \quad (1)$$

The SAOD is computed as the total Mie backscatter vertically integrated between the zonally-averaged tropopause altitude $Z_{trop}$+1 km and 30 km and multiplied by a constant lidar ratio (LR) of 50 sr:

$$SAOD = LR \times \int_{Ztrop+1}^{30}(Mie_{copolar} + Mie_{crosspolar}). \quad (2)$$

The thermal tropopause altitude is obtained from European Center of Medium-range Forecasts (ECMWF) meteorological data provided with L2A product (Donovan et al., 2024). The 1 km shift above the tropopause is applied in order to exclude clouds reaching the tropopause altitudes.

**2.2   Ground-based lidars**

The ground-based lidars involved in this study are affiliated to the Network for Detection of Atmospheric Composition Change (NDACC) network (www.ndacc.org) and are subject to regular operation with a typical rate of 2-5 measurement nights per week.

### 2.2.1   Maïdo observatory

The Maïdo station, a high-altitude facility of the Atmospheric Physics Observatory of La Réunion (OPAR), is located in the South-West Indian Ocean (21°S, 55°E, 2160 m a.s.l.) on the west side of Reunion island. The facility (Baray et al., 2013) hosts since 2013 the LiO3S lidar operating at 355 nm routinely with an average of 8 acquisition nights per month. The aerosol optical properties are retrieved at a height-dependent vertical resolution varying between 30 and 180 m below 30 km, and using the two-component Klett inversion method with a prior knowledge of i) the molecular backscatter profiles calculated

with atmospheric pressure and temperature from ECMWF, ii) the aerosol optical depth assessed with the Rayleigh slope method in the stratosphere and iii) a constant LR of 50 sr. A more detailed description of the instrument, the aerosol retrieval method and the error budget are provided in Gantois et al. (2024) and references therein.

### 2.2.2   Table Mountain Facility (TMF)

The JPL Table Mountain Facility, California (34.38∘ N, 117.68∘ W; 2285 m a.s.l.) is located in the San Gabriel Mountains, north of the LA Basin. The site hosts numerous instruments for atmospheric composition monitoring, including a stratospheric ozone differential absorption lidar (TMSOL) from which the aerosol data used here are obtained, a tropospheric ozone lidar (TMTOL), a water vapor Raman lidar (TMWAL), which have been providing multi-decadal long-term vertical profiles of ozone, temperature, aerosol and water vapor for NDACC and for the Tropospheric Ozone Lidar Network (TOLNet). The TMF

data used here were obtained from the nighttime-only, high-intensity UV Rayleigh (355 nm) and Raman (387 nm) lidar returns of the TMSOL instrument measured two- to five-times per week. The instrumental configuration since 2003 uses a mix of the instrument's excimer-based original design [McDermid et al., 1991] and a newer Nd:YAG-laser-based design [McDermid et al., 1995]. The data used here were processed using the Global Lidar Analysis Software Suite (GLASS) v1 algorithm, which includes NDACC-standardized vertical resolution and uncertainty budget [Leblanc et al., 2016a, 2016b]. The measured

quantity is the backscatter ratio at 355 nm normalized at 32-36 km altitude, with a nearly-constant total uncertainty of 10% throughout most of the profile, yielding a vertical resolution ranging between a few tens of metres in the lowermost stratosphere to 2.5 km in the upper stratosphere. The backscatter coefficient is derived from the backscatter ratio using the 3-hourly atmospheric density output from MERRA-2 interpolated at the site. The profiles are archived at NDACC in HDF format and publicly available about 1-2 months after measurement. A Rapid Delivery (RD) version is also available 2 days after

measurement.

### 2.2.3   Observatoire de Haute-Provence (OHP)

The Observatoire de Haute-Provence (OHP) located in southern France (43.9° N, 5.7° E, 670 m) is equipped with several lidar systems for atmospheric sounding at a wide range of altitudes. Here we use the aerosol measurements by LiO3S

lidar operating at 355 nm on a regular basis with a mean measurement rate of 10-12 acquisition nights per month (Godin-Beekmann, 2003). The stratospheric aerosol data from this lidar are available since 1994. For retrieving vertical profiles



of stratospheric aerosol, we apply Sasano-Fernald inversion method, which provides backscatter and extinction coefficients. The scattering ratio is then computed as a ratio of total (molecular plus aerosol) to molecular backscattering, where the latter is derived from ECMWF meteorological analysis. The resulting vertical profiles of aerosol parameters are reported at 15 m
vertical resolution. A more detailed description of the instruments, aerosol retrieval and error budget are provided by Khaykin et al. (2017) and references therein.

### 2.2.4 Lauder observatory

The Lauder Atmospheric Research Station is located on the South Island of New Zealand (45.0° S, 169.7° E, 370 m). The lidar has been operating at 532 nm since 1992 on a regular basis, with a mean measurement rate of 1 or 2 nights per week
before February 2023 and continuously at 10-minute intervals after that. It has a capability of both scattering ratio and particle depolarization ratio measurements. For retrieving the vertical profile of the scattering ratio, we apply Fernald inversion method assuming the aerosol extinction-to-backscatter ratio varying with height between 33 sr and 58 sr according to the balloon-borne optical particle counter (OPC) measurements over Lauder (Sakai et al. 2025). The molecular backscattering coefficient is derived from the JRA-55 meteorological analysis. The resulting vertical profiles of aerosol parameters are reported at 150
m vertical resolution. A more detailed description of the instruments, aerosol retrieval and error budget are provided by Nagai et al. (2010) and Sakai et al. (2016, 2025). For conversion of the 532 nm backscatter to 355 nm, a backscatter Angstrom exponent (BAE) of 1.8 was used (Chouza et al., 2020).

### 2.3 AZOR balloon-borne backscatter sonde

The AZOR balloon-borne backscatter sonde measures atmospheric aerosols by detecting light scattered from short-range
(0.2 – 5 m) volumes (~0.1 m³) illuminated by LED pulses at 528 nm and 940 nm. Using a lens-based photodiode detector system at ~170°–180° scattering angles and a 5° offset between emitter and detector axes, it achieves high signal-to-noise ratios (≥50 at 30 km) via synchronous detection (Balugin et al., 2022). Designed for night-time operation, the sonde weighs less than 1 kg (including batteries) and can be deployed on small weather balloons either in combination with standard radiosondes or in fully autonomous mode, with onboard navigation, telemetry, and data logging systems enabling versatile use
in remote locations and under a variety of atmospheric research scenarios, such as monitoring of polar stratospheric clouds, tropospheric and stratospheric aerosols, cirrus, wildfire and volcanic plumes as well as for validation of remote ground- and satellite-based aerosol observations.

Balloon-borne measurements using the AZOR sonde, complemented by additional aerosol sensors, were conducted from Orléans, France (47.8° N, 1.9° E), with launch times coordinated to coincide with nighttime ATLID overpasses. To enable a
direct comparison with ATLID observations, AZOR backscatter profiles at 528 nm and 940 nm were used to derive the backscatter Ångström exponent. This exponent was then applied to convert the 528 nm backscatter to 355 nm, to match the ATLID operating wavelength.

### 2.4 ATLID collocations with suborbital measurements

As a collocation criterion for comparison of ATLID with suborbital measurements, we used a maximum distance of 100 km and a time window of ±12 hours. Most of the ATLID collocations presented in this study lay within 60 km and 6 hours. All the collocated measurements were conducted during nighttime. It should be noted that for the stratospheric intercomparison, the collocation criteria are less critical than for highly variable tropospheric cloud scenes. For the ground-based lidars, we used standard-issue data (accessible via the NDACC database) obtained during nighttime acquisitions lasting
2–4 hours. For the intercomparison setup, ATLID L1B data were averaged over a 10 s time interval, corresponding to an along-track distance of ~75 km or ~510 laser shots. The resulting scattering ratio profiles were interpolated onto a fixed vertical grid of 100 m and smoothed using a 5-point boxcar filter.

### 2.5 Extinction measurements by NOAA-21 OMPS-LP

The OMPS Limb Profiler (LP) aboard NOAA-21, launched in November 2022 as the second instrument in the OMPS
LP series with first data products available in February 2023, measures limb-scattered solar radiation across UV, visible, and near-infrared wavelengths (290-1000 nm) to retrieve vertical profiles of ozone density and aerosol extinction coefficients in the lower stratosphere to the upper stratosphere. The instrument employs three parallel vertical slits with a backward-looking geometry along the orbit track, providing cross-track separation of approximately 250 km at the tangent point and vertical sampling at 1 km altitude intervals up to 80 km with 1.8 km vertical resolution. A significant operational change occurred
on December 12, 2023, when the integration time was reduced from 15 seconds to approximately 7 seconds, improving along-track sampling to 44 km. The version 1.0 retrieval algorithm provides the aerosol extinction profiles at six independent wavelengths, 510, 600, 675, 745, 869, and 997 nm (Taha et al., 2021).



### 2.6 ISS SAGE III

The Stratospheric Aerosol and Gas Experiment (SAGE) III/ISS provides stratospheric aerosol extinction coefficient profiles using solar occultation observations from the International Space Station (ISS) (Cisewski et al., 2014). These measurements, available since February 2017, are provided for nine wavelength bands from 384 to 1543 nm and have a vertical resolution of approximately 0.7 km. Here we use version V6.0 of SAGE III solar occultation species data at 384 nm converted to 355 nm to match the ATLID wavelength using the Angstrom exponent derived from SAGE III multiwavelength extinction data. For comparison with OMPS-LP extinction at 869 nm, we use directly the 869 nm channel of SAGE III. The SAOD is computed by integrating aerosol extinction in the same altitude range as for ATLID (Sect. 2.1).

### 2.7 GloSSAC

The Global Space-based Stratospheric Aerosol Climatology (GloSSAC) is a 38-year climatology of stratospheric aerosol extinction coefficient measurements by various satellite instruments such as SAGE, OSIRIS, CALIOP (Kovilakam et al., 2020). Data from other space instruments and from ground-based, aircraft and balloon-borne instruments are used to fill in key gaps in the data set. Here we use GloSSAC V2.2 data on aerosol extinction at 525 nm available until the end of 2023. The SAOD is computed by integrating aerosol extinction in the same altitude range as for the other satellite data sets.

### 2.8 $SO_2$ and AAI observations with Sentinel-5p TROPOMI

The Tropospheric Monitoring Instrument (TROPOMI) is a nadir-viewing hyperspectral imaging spectrometer aboard ESA's Sentinel-5 Precursor satellite, launched in 2017. It measures backscattered solar radiation in the ultraviolet, visible, near-infrared, and shortwave infrared spectral ranges, enabling retrievals of key atmospheric trace gases and aerosols at high spatial resolution (up to $3.5 \times 5.5$ km²) (Veefkind et al., 2012). Here, we use the offline (OFFL) version of the Sentinel-5P TROPOMI sulfur dioxide ($SO_2$) and absorbing aerosol index (AAI) Level-2 products. $SO_2$ retrievals are based on Differential Optical Absorption Spectroscopy (DOAS) analysis of ultraviolet spectra and provide vertical column densities for different assumed plume heights (Theys et al., 2017). AAI is derived from ultraviolet spectral bands (340–380 nm) (Veefkind et al., 2012) and calculated using the spectral contrast between a pair of UV wavelengths, based on the ratio of the observed top-of-atmosphere reflectance and a pre-calculated theoretical reflectance for a Rayleigh-scattering-only atmosphere. Positive AAI values indicate the presence of UV-absorbing aerosols, such as ash, dust and smoke. AAI is influenced by aerosol properties, including optical thickness, single scattering albedo, as well as the aerosol layer height. In this study, we use this parameter only as a general indication of the presence of ash in the eruptive column.

## 3. Results

### 3.1 Ruang eruption sequence in April 2024

In April 2024, after ~22 years of dormancy, Indonesia's Ruang volcano (2.31°N, 125.37°E) experienced a significant eruptive sequence, characterized by two main explosive phases that injected volcanic aerosol directly into the stratosphere. Following initial seismic unrest, the most intense eruptive pulses occurred between 16–17 and 29–30 April. The Volcanic Explosivity Index (VEI) assigned to the Mount Ruang eruption in April 2024 is VEI 4 (Global Volcanism Program, 2025).

On 17 April, two major eruptive pulses at approximately 10:00 and 12:15 UTC generated sub-Plinian columns that reached stratospheric altitudes. The eruptive $SO_2$ cloud moving westward has been observed on the next day at 05:00 UTC by TROPOMI instrument (Fig. 1a) with the peak $SO_2$ columnar values of 178 DU and the stratospheric $SO_2$ mass estimated at 0.25 Tg (S. Carn, personal communication). At about the same time (06:00 UTC), the eruptive plume was sliced by OMPS-Limb Profiler (LP) instruments onboard the NOAA-21 platform; its ground projections are shown as grey circles in Fig. 1a. The vertical section of the OMPS-LP aerosol Extinction Ratio (ER, ratio of aerosol to molecular extinction) along the A – B orbital segment reveals an intense plume above the tropopause with ER at 869 nm reaching 91 (Fig. 1b). While the core of the plume was bounded within the 20-23 km altitude layer, its northern part extended up to 25 km, i.e. the maximum altitude reached by the eruptive column.

A second, similarly intense eruptive phase occurred early on 29 April at approximately 17:30 UTC. The $SO_2$ image from TROPOMI (Fig. 1c) acquired on the next date at around 05:00 UTC (i.e. 11 hours after the eruptive phase) puts the maximum $SO_2$ columnar amount at 810 DU, that is significantly larger compared to the first eruptive phase on 17 April. Nevertheless, the mass of the stratospheric $SO_2$ plume was estimated at 0.16 Tg. An enhancement of UV Absorbing Aerosol Index peaking at 10 was also detected by TROPOMI within the $SO_2$ cloud (black contours in Fig. 1c), which indicates the presence of absorbing ash particles in the eruptive plume. The ash particles must have been rapidly removed from the plume through sedimentation as the TROPOMI observations on the next day (not shown) do not indicate any AAI enhancement. The OMPS-LP vertical section across the western boundary of the plume (Fig. 1d) shows the core of the plume at ~21.5 km altitude with the maximum ER of 107 and the top of the cloud at ~24 km.



The volcanic plumes from both eruptive phases were carried eastward by the tropical stratospheric westerlies whereas the upper parts of the plumes were moving faster due to the vertical wind shear. The aerosol plume from the first eruptive phase at ~24 km altitude has completed its first circumnavigation by 5 May, that is in 3 weeks.

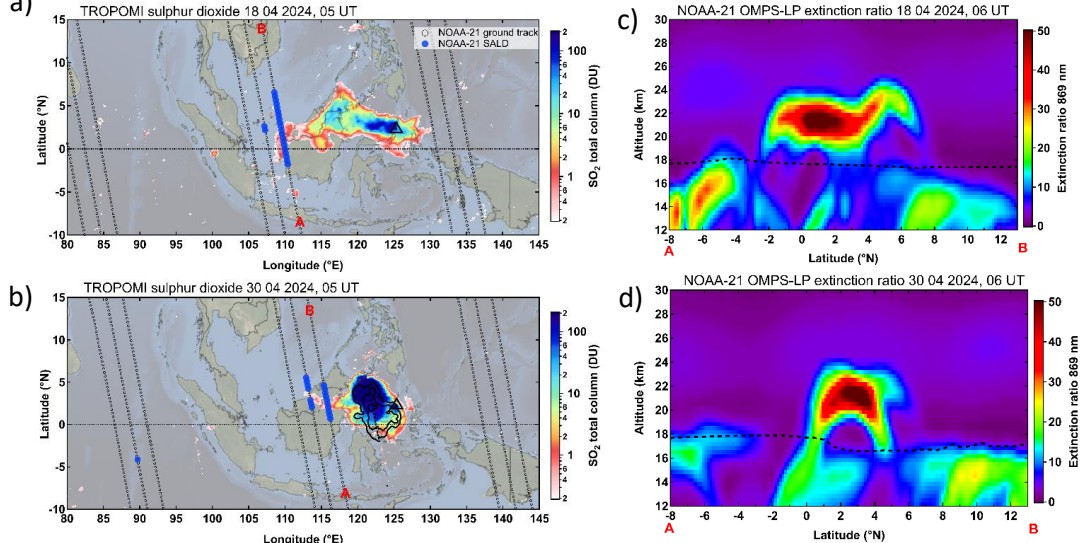

**Figure 1. Ruang eruptive sequence in April 2024. a) SO2 total column map from TROPOMI on 18 April 2024 overlaid by NOAA-21 OMPS-LP ground projections (open circles) and stratospheric aerosol layer detections (SALD, blue circles). The location of Ruang volcano is shown as triangle. b) Same as a) but on 30 April 2024 with black contours indicating TROPOMI absorbing aerosol index above 2 with contour interval of 3. c) Vertical section of OMPS-LP aerosol extinction ratio at 869 nm along the orbital segment A – B (as indicated in the map) on 18 April 2024. Dashed curve marks the tropopause altitude. d) same as c) but on 20 April 2024.**

## 3.2  Global evolution of stratospheric aerosol perturbation

By 17 May 2024, one month after the first eruption, the Ruang aerosols have attained a complete zonal spread and the aerosol enhancements could be observed by OMPS-LP at any longitude within the 10° S -10° N latitude belt. The subsequent meridional dispersion of sulphate aerosols can be inferred from Figure 2, which shows bi-daily zonal averages of stratospheric aerosol optical depth (SAOD) as a function of time and latitude from OMPS-LP and ATLID overlaid by SAGE III observations of SAOD at the corresponding wavelengths shown as color-coded circles. By mid-June, the SAOD perturbation extended across the tropical belt and started propagating into the southern extratropics leading to enhanced SAOD throughout the southern hemisphere by October 2024, i.e. by the end of the Antarctic vortex season. With that, the bulk of aerosol perturbation has remained within the tropics.

A coherent picture is observed by ATLID (Fig. 2b), whose observations provide near global coverage including the polar night regions and thereby capturing the Polar Stratospheric Clouds (PSC) signal in the Arctic and Antarctica. It should be noted that both nighttime and daytime ATLID data have been used for analysis (Sect. 2.1). Available since mid-August 2024, ATLID observations show the bulk of Ruang aerosols in the tropics and the secondary maximum in the southern mid-latitudes, in consistency with OMPS-LP and SAGE III data. The absolute values of SAOD at 355 nm are in fair-to-good agreement with SAGE III data considered as reference for aerosol extinction. A detailed quantitative comparison of ATLID L1/L2 products with SAGE III will be the subject of a future study.

In November 2024, as the northern stratosphere started to turn to the winter regime, enabling a more efficient transport from the tropical belt towards the extratropics, the SAOD perturbation has gradually extended into the northern midlatitudes thereby obtaining nearly complete meridional coverage except the Arctic region dynamically isolated by the polar vortex. Both OMPS-LP and ATLID suggest that by early May 2025 the northern extratropical SAOD has returned to the pre-eruption levels, whereas the southern extratropics were still polluted by Ruang sulphate aerosols, which were bound to be entrained by the 2025 Antarctic polar vortex.

In May 2025, the tropical SAOD perturbation started to migrate towards the northern tropics and this process – likely related to the change of the QBO phase around that time - is equally well captured by ATLID and OMPS-LP. A noticeable SAOD perturbation in the northern extratropics during June-July is associated with a major outbreak of wildfires and pyroCb activity in Canada and Siberia (Panboreal Wildfire Outbreak, PWO) in late May 2025. The stratospheric impact of this event will be characterized by another ongoing study.



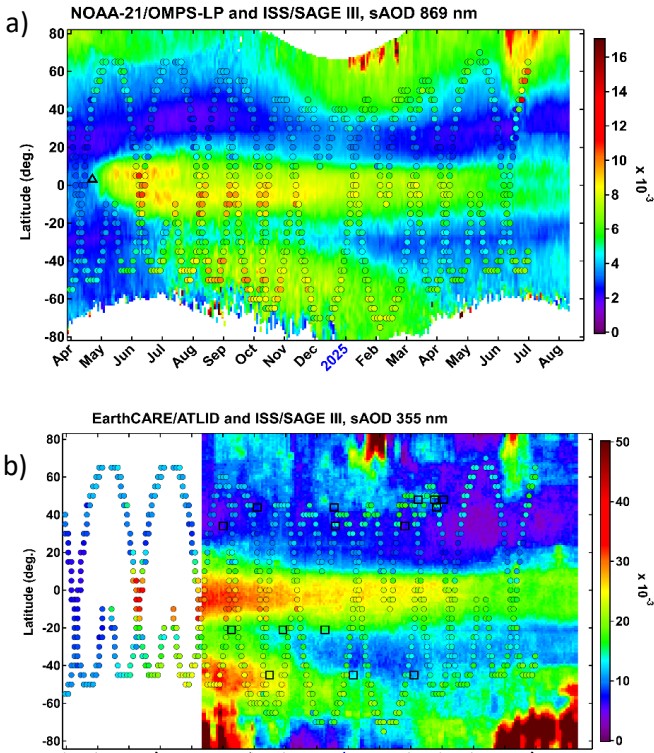

**Figure 2. Time-latitude variation of zonal-mean stratospheric aerosol optical depth (SAOD) since April 2024 from OMPS-LP at 869 nm (a) and ATLID at 355 nm (b). Circles in both panels indicate SAGE III 3 day/5° latitude averages of SAOD at the corresponding wavelengths. Black triangle in a) marks the location/timing of Ruang eruption. Black squares in b) indicate the timing and location of suborbital measurements with NDACC lidars and AZOR balloon soundings.**

The evolution of Ruang aerosols in the latitude-altitude dimension from OMPS-LP and ATLID is shown in Fig. 3. In September 2024 (Fig. 3a,b), the bulk of Ruang aerosol occupied the tropical belt between 19 – 23 km altitude, whereas the enhanced aerosol signal extended to higher altitudes in the upwelling branch of Brewer-Dobson circulation, as well as to the southern midlatitudes, as a result of the poleward isentropic transport along the 450 K isentrope. The poleward propagation of
305 Ruang sulphates was limited to ~60° S by the mixing barrier of the Antarctic polar vortex in September 2024, whereas the aerosol enhancement in the lowermost stratosphere south of 70° S can be associated with PSC particles.

By January 2025 (Fig. 3c,d), the Ruang aerosols in the tropics have ascended by about 1-2 km with the average rate of 7.3 m day$^{-1}$. Meanwhile, the southern wing of aerosol perturbation has extended all the way to the South pole along the isentropic layer between 400 – 550 K. A similar pattern can be seen in the Northern extratropics, where the further poleward
dispersion was disabled by the mixing barrier at the Arctic vortex, which exhibits a strong PSC signal reaching unusually high altitudes for the Arctic with respect to the climatology (Tritscher et al., 2021). Three months later, by April 2025, the Ruang aerosol have further ascended in the tropics, whereas the extratropical wings have weakened and subsided to lower altitudes, as can be inferred from Fig. 3e,f. One should also note the northbound extension of the tropical aerosol perturbation around the 750 K isentropic surface (~28 km) during that period, which may be linked with the decay phase of the Arctic vortex,
enhancing the poleward transport along the deep branch of Brewer-Dobson circulation, as was pointed out by Khaykin et al. (2022b) regarding the transport of the Hunga aerosols.

In general, the evolution of Ruang's aerosols in the latitude-altitude dimension is consistently resolved by OMPS-LP and ATLID instruments as far as the tropics and southern extratropics are concerned. However, in the northern extratropics ATLID shows reduced sensitivity to the volcanic aerosol layer, for which the reason remains to be investigated.





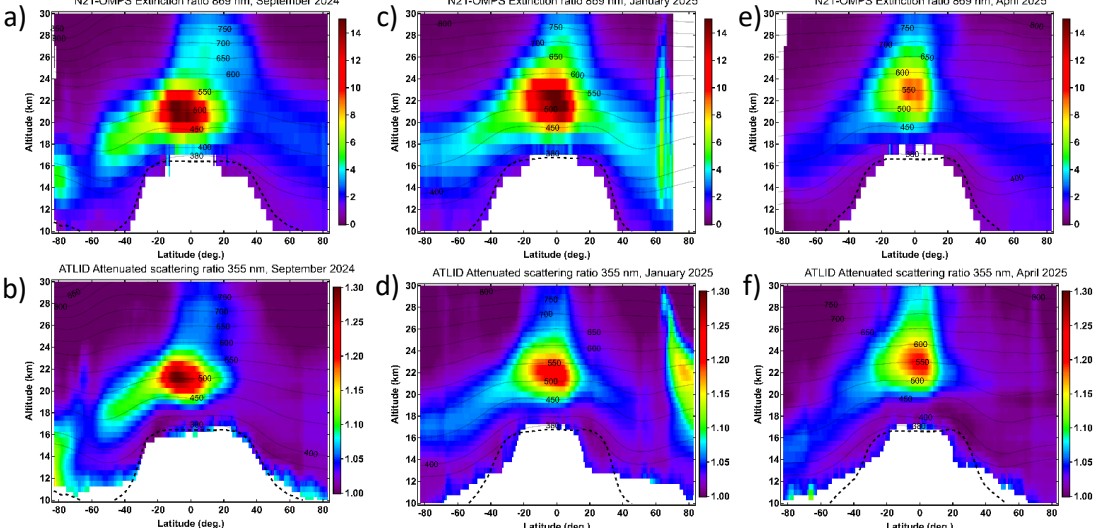

**Figure 3. Zonal/monthly-mean latitude-altitude distribution of OMPS-LP extinction ratio at 869 nm (a,c,e) and ATLID scattering ratio at 355 nm (b,d,f) for September 2024 (a,b), January 2025(c,d) and April 2025 (e,f). Dashed curve indicates the tropopause altitude, solid contours mark potential temperature levels.**

### 3.3 ATLID and ground-based lidars

In the previous section, we provided a global perspective on the transport of Ruang aerosols and pointed out a good qualitative agreement between ATLID and limb viewing observations by OMPS-LP and SAGE III. In this section, we assess the ATLID stratospheric performance through comparison with NDACC lidars at different locations, spanning 45° S to 48° N, and analyse the global progression of Ruang aerosols from the combined satellite and ground-based perspectives.

Figure 4 combines the ATLID and NDACC lidar observations in the Southern hemisphere. The evolution of zonal-mean ATLID scattering ratio in the tropical and mid-latitude bands, corresponding to the Maido station at La Reunion (21° S) and Lauder station in New Zealand (45° S) is shown in Fig. 4a,e (top panels). The dates of the selected collocated lidar measurements at each station, representing different stages of the Ruang aerosols evolution, are indicated as vertical dashed lines. The positions of these collocations in the time-latitude dimension are shown in Fig. 2b.

The first collocated measurement with ATLID at the Maido tropical station took place on 9 September 2024 (Fig.4b) when both lidars reported a broad aerosol layer extending between 19 – 23 km, and reflecting the zonal-mean distribution of Ruang aerosols at this time stage shown as green dashed curve. The next collocation on 29 October (Fig. 4c) displays a similar vertical structure of the aerosol profile peaking at 20 km altitude, which is somewhat different from the zonal-mean profile, characterized by a weaker peak at ~21 km. Another collocation on 9 December (Fig. 4d) captures a triple-peak structure of the aerosol enhancement, vastly different from the zonal-mean profile, suggesting active regional transport of Ruang aerosols from the tropical belt to the subtropics. One should note an excellent agreement between the collocated lidar measurements consistently resolving the fine vertical structure of stratospheric aerosol layers.

At the southern midlatitudes, the Ruang aerosol layer has been observed since the beginning of ATLID observations, between 15 – 22 km altitude with the maximum intensity at ~18.5 km (Fig. 4e). The ATLID measurement collocated with Lauder lidar on 16 October 2024 (Fig. 4f) reveals a particular scene with a sharp aerosol layer near 20 km altitude, notably exceeding the zonal-mean ATLID profile. A similar picture was observed on 6 January 2025: the twin-peak layer appears stronger over Lauder compared to the zonal-mean ATLID profile. Later, on 6 March 2025, both satellite and ground-based lidars consistently report a higher-altitude peak at 22 km, which may be associated with in-mixing of the tropical Ruang aerosols after their diabatic ascent in the BDC upwelling branch. As in the previous cases, the local profiles exhibit somewhat lower scattering ratios compared to the zonal-mean profile, although the double-peak structure is evident in both local and zonal-mean profiles.

The agreement between ATLID and Lauder lidar is generally very good, especially in terms of the fine vertical structure of the aerosol enhancement in the lower/middle stratosphere. The upper part of ATLID profiles, above ~25 km tends to be noisier and prone to noticeable deviations with respect to the ground-based data within 1-2 km-thick layers.



**Figure 4. Intercomparison of ATLID and NDACC lidar observations in the Southern hemisphere. Top row panels display ATLID time series of zonal-mean scattering ratio profile within the 4-degree latitude bands 23° S – 19° S (a) and 47° S – 43° S (e) corresponding to Maido and Lauder stations. Dashed curves indicate the tropopause altitude, vertical dashed lines indicated the timing of selected collocated measurements. The results of intercomparison are shown in panels b), c) and d) for Maido station and in f), g) and h) for Lauder station. Horizontal dashed line marks the local tropopause altitude. The dates of collocations are provided in each panel.**



**Figure 5. Intercomparison of ATLID, NDACC lidar observations and AZOR balloon soundings in the Northern hemisphere. Top row panels display ATLID time series of zonal-mean scattering ratio profile within the 4-degree latitude bands 32° N – 36° N corresponding to the TMF station (a), 42° N – 46° N corresponding to OHP station (e) and 46° N – 50° N corresponding to AZOR sounding locations (i). Dashed curves indicate the tropopause altitude, vertical dashed lines indicated the timing of selected collocated measurements. The results of intercomparison with TMF and OHP lidars are shown in panels b), c) and d) and in f), g) and h) respectively. The results of intercomparison with AZOR balloon sonde are shown in panels j), k) and l). Horizontal dashed line marks the local tropopause altitude. The dates of collocations are provided in each panel.**



The transport of Ruang aerosols into the northern extratropics was analysed using lidar collocations at Table Mountain Facility (TMF, 34° N) and Observatoire de Haute Provence (OHP, 43.9° N). The evolution of zonal-mean vertical profile of ATLID scattering ratio (Fig. 5a) suggests the arrival of Ruang aerosols to the northern subtropics in October 2024. The first collocated measurement at TMF on 1 September (Fig. 5b) shows background aerosol conditions except for a subtle enhancement around 26 km altitude that could be associated with the remnants of aerosols produced by the Hunga eruption in

January 2022, which were still present in the stratosphere by that time (Khaykin et al., 2025). The later collocation on 19 December (Fig. 5c) reveals a pronounced layer of enhanced scattering ratio that can undoubtedly be linked with Ruang aerosols. The enhancement magnitude in the local measurements exceeds that of the zonal-mean profile, indicating that the aerosol perturbation has not attained the zonal uniformity at the subtropical latitudes by that time. As a result of the further dispersion of Ruang aerosols across the northern extratropics, the TMF collocation on 25 February exhibits a weaker

enhancement, reflecting the zonal-mean state (Fig. 5d).

    A similar development of aerosol profile is observed at OHP latitudes (Fig. 5e), although with a lag of about one month, representative of the poleward transport timescale. The 4 October's ATLID collocation at OHP reflects the near-background aerosol condition (Fig. 5f). In mid-December 2024, both ATLID and OHP lidar report two broad aerosol enhancements centred at ~19 km and ~24 km (Fig. 5g). While the lower one can be associated with Ruang aerosols, the upper one may possibly be

linked with the remnants of Hunga aerosols. The later OHP collocation on 27 March (Fig. 5h) captures a scene with a pronounced boundary between the Ruang aerosol layer and aerosol-free stratosphere above ~23 km altitude. While the lower part of the Ruang aerosol layer is zonally uniform, as suggested by the zonal-mean profile, the upper part of the enhanced layer at 20 – 22 km, deviating from zonal-mean profile, appears to be a regional feature.

**3.4 Suborbital balloon-borne measurements**

    An additional effort towards the assessment of ATLID stratospheric performance was realized through a series of small balloon launches near Orleans, France (48° N) coordinated with ATLID collocated overpasses and carrying various aerosol sensor, including the AZOR backscatter sonde. The nature of in situ measurements is different from the ground-based lidar acquisition lasting several hours, and the balloon soundings provide a quasi-instantaneous view of the atmosphere, similarly

to ATLID sampling.

    The ATLID-collocated balloon soundings with AZOR sonde started in March 2025, when the Ruang stratospheric aerosol layer was fading out in the northern midlatitudes (Fig. 5i). The sounding of 10 March (Fig. 5j) reports a subtle yet distinct layer at 19 km, resolved coherently by ATLID and AZOR. By 26 March (Fig. 5k), the aerosol layer obtains sharper boundaries, while preserving its magnitude. The subsequent sounding on 4 April (Fig. 5l) shows a complex vertical structure of the

scattering ratio profile, which is remarkably well reproduced by satellite and balloon-borne observations. While the accumulation of aerosol layers in the 18 - 22 km layer might be linked with the dissipation of Ruang aerosols, the origin of the upper enhancement around 26 km may be the result of the poleward transport of the volcanic aerosols from the tropics along the 750 K isentropic surface (Fig. 3e,f).

**3.5 Ruang perturbation in a long-term perspective**

    Figure 6 displays SAOD time series since the beginning of the XXI century for the southern and northern midlatitudes as well as for the tropical band from GloSSAC merged satellite dataset available until the end of 2023 and extended by ISS/SAGE III observations since 2024. The high-latitudes are excluded from the analysis due to limitations of SAGE III latitude coverage. The timing of volcanic eruptions and wildfire outbreaks that led to a measurable increase of the

zonal/monthly-mean SAOD is indicated in Fig. 6 as triangles and stars coloured according to the latitude band of the event. Eighteen such events can be identified since the year 2000, including eleven tropical eruptions, four extratropical eruptions and three wildfire outbreaks. The three largest perturbations were produced by the Hunga eruption in 2022 (Khaykin et al., 2022b), the Australian New Year (ANY) wildfire super outbreak (Khaykin et al., 2020; Peterson et al., 2021) in 2019/2020 and the Raikoke eruption in 2019 (Kloss et al., 2021; Khaykin et al., 2022a) that occurred at about the same time with the

tropical Ulawun eruptions (Kloss et al., 2021). The SAOD perturbation produced by the 2024 Ruang eruption is on par with those produced by other significant events (2009 Sarychev, 2011 Nabro and 2015 Calbuco eruptions) and represents the second largest tropical perturbation after Hunga in the post-Pinatubo era, that is after the June 1991 eruption of Mt. Pinatubo.

    As most of the other tropical eruptions (and particularly those in the southern tropics, e.g. Hunga, Aoba, Kelud), Ruang eruption produced a hemispherically-asymmetric perturbation with much larger impact on the southern hemisphere as can be

concluded by comparing the different curves in Fig. 6. Furthermore, in the case of Ruang (located in the northern tropics), the magnitude of the southern midlatitude perturbation is close to its tropical counterpart. The Ruang's impact on the northern midlatitudes, maximizing in early 2025, appears to be relatively large for a tropical eruption and exceeds the Hunga-induced SAOD enhancement in the Northern hemisphere around the turn of the year 2023.






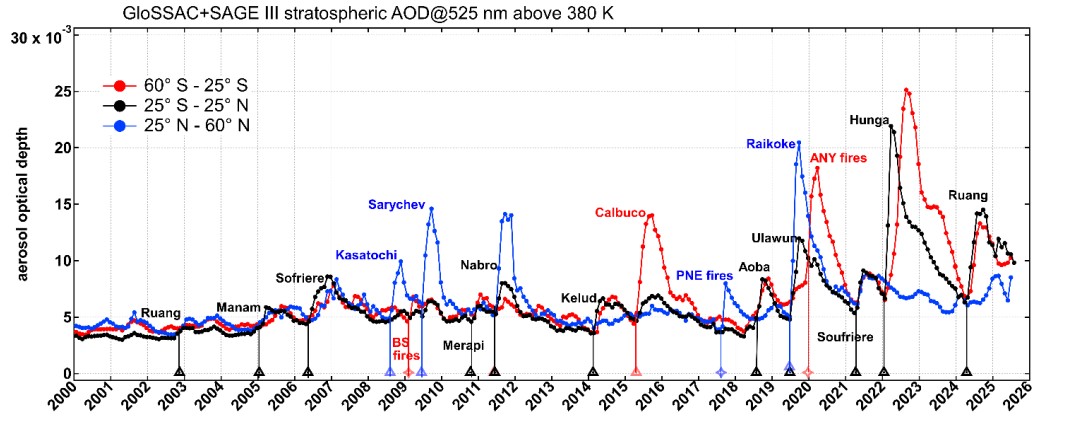

**Figure 6. Time series of stratospheric aerosol optical depth (SAOD) at 525 nm from GloSSAC merged satellite record and SAGE III since January 2024 for different latitude bands. Triangles and stars along the time axis mark the timing of respectively volcanic eruptions and wildfire events that produced measurable increase of the zonal/monthly-mean SAOD. The marker color correspond to the latitude band of the event.**

### 3.6 Discussion and summary

During volcanically-quiescent period that followed the full decay of Pinatubo aerosols by late 1997, the stratospheric aerosol loading remained at background levels until 2003 (Vernier at al., 2011; Kremser et al., 2016; Khaykin et al., 2017). This period was terminated by the VEI 4 eruption of Ruang in September 2002, which injected a relatively small amount of $SO_2$ into the stratosphere causing a subtle SAOD perturbation (Kremser et al., 2016), much smaller than the 2024 eruption. The subsequent series of tropical eruptions led to a systematic increase of stratospheric aerosol loading (Vernier at al., 2011), which was initially misattributed to the increase of anthropogenic $SO_2$ emissions (Hofmann et al., 2009). The volcanic activity in the Northern hemisphere during 2008 – 2011 resulted in several distinct SAOD perturbations, which was followed by a quiescent period that lasted until the Kelud eruption in early 2014 (Vernier et al., 2016). Since 2017, the significant SAOD perturbations, caused by either volcanic eruptions or wildfire outbreaks in both hemispheres, occurred at least once per year, maintaining the global stratospheric aerosol loading well above the background levels.

The 2024 Ruang eruption occurred during the late decay phase of the Hunga stratospheric aerosol perturbation (Khaykin et al., 2025). The eruptive sequence during the second half of April 2024 injected a total of ~0.4 Tg of $SO_2$ into the stratosphere. The eruptive column reached altitudes of 19-25 km, which is notably high for a VEI 4 eruption (Kremser et al., 2016). For comparison, the $SO_2$ mass injected by the VEI 5 Hunga eruption was estimated at ~0.5 – 1 Tg (Carn et al., 2022; Sellitto et al., 2024), whereas its young aerosol plume extended vertically between 20 – 30 km altitude (Taha et al., 2022; Khaykin et al., 2022b). The Ruang eruptions caused a doubling of the tropical SAOD and a factor of 1.8 increase of the southern extratropical SAOD for several months. By the time of writing (August 2025), the SAOD has not yet returned to the pre-eruption levels, suggesting the lifetime of Ruang aerosol in the stratosphere of at least 16 months. The magnitude and duration of SAOD perturbation by the Ruang eruption ranks it among the strongest volcanic events in the post-Pinatubo era despite a relatively modest mass of injected $SO_2$ compared to other VEI 4 eruptions. The sulfate aerosols observed in the Ruang plume by Atmospheric Chemistry Experiment (ACE) were about 64% (by weight) sulfuric acid, and the particles had an average median radius of 0.127 μm (Dodangodage et al., 2025). Silicate features from volcanic ash were observed by ACE only in the very young plume and no indications for enhanced $H_2O$ mixing ratios were reported.

The injection altitude of Ruang eruption reaching the middle stratosphere has led to a more efficient poleward transport of volcanic aerosols into the winter hemispheres, as was previously observed after the Hunga eruption (Khaykin et al., 2022b, Legras et al., 2022). With that, the timing of Ruang eruption (that occurred during the Austral Fall season) led to an expedited onset of the poleward transport to the southern extratropics during the Boreal winter 2024, which is reflected by the shorter lag between the tropical and the southern extratropical SAOD perturbations for Ruang as compared to that for Hunga (Fig. 6). Despite the early onset of southbound transport, the Ruang sulphates arrived to the southern extratropics by the time the 2024 Antarctic vortex was already fully formed, effectively disabling the further poleward propagation of Ruang aerosols across the dynamical vortex boundary. However, by the time of the 2025 Antarctic vortex onset, the entire Southern hemisphere was polluted by Ruang sulphates (Fig. 2) and these aerosols have obviously been entrained by the 2025 vortex. The possible impact of the Ruang sulphates on the 2025 Antarctic PSC occurrence and ozone hole (as was reported for Calbuco eruption by Stone et al. (2017)) remains to be investigated.



While the transport of Ruang aerosols towards the northern extratropics was less efficient during Boreal Summer 2024, a significant influx of aerosols into the northern midlatitudes has been observed by satellite and ground-based instruments during Boreal winter 2024/2025. In Spring 2025, the ground-based lidars and in situ aerosols sondes reported distinct stratospheric layers (Figs. 4 and 5) in consistency with satellite observations of Ruang aerosols' meridional progression.

The 2024 Ruang eruption occurred shortly before the launch of ESA's EarthCARE ATLID mission and provided an excellent natural testbed for assessment of ATLID's stratospheric performance using limb-viewing satellites and the NDACC lidar network spanning a wide latitude range. While there exists only a few stratospheric aerosol lidars systems operating at 355 nm on a regular basis, here we have employed most of them for direct comparison with ATLID collocated measurements. The intercomparison indicates excellent agreement between the NDACC lidars at 4 different locations in both hemispheres. The vertical structure of aerosol enhancements produced by Ruang and its evolution driven by meridional progression of volcanic aerosols are consistently resolved by ATLID and ground-based lidars. One should note that ATLID is the only existing satellite instrument whose demonstrated performance in stratospheric aerosol profiling closely approaches that of ground-based stratospheric lidar systems in terms of the vertical resolution and signal-to-noise ratio. Indeed, with just a 10-second acquisition, ATLID proved capable of resolving fine vertical structures only a few hundred meters thick— readily comparable to what ground-based lidars achieve after several hours of backscatter signal accumulation.

ATLID's observations of the global progression of Ruang aerosols align closely with those from OMPS-LP and SAGE III, whilst uniquely providing coverage through polar night. While the assessment of ATLID Level-2 products in the stratosphere has yet to be carried out, the Level-1B product exploited in this study is herein qualified for a broad range of scientific applications concerning stratospheric aerosols and their modulation by volcanism, wildfires, and anthropogenic emissions.

### Acknowledgements

We thank the personnel of lidar stations (OHP, Maido, TMF and Lauder) for conducting lidar measurements. The French lidar stations operate under support of French Institut National des Sciences de l'Univers (INSU) of the Centre National de la Recherche Scientifique (CNRS) and of Centre National d'Etudes Spatiales (CNES).

### Funding

Agence Nationale de la Recherche (ANR) PyroStrat 21-CE01- 335 0007-01 project (SK, SGB, NK)
CNRS INSU CPJ STANDARDS project (SK, NK)
CNES EXTRA-SAT (SK, SGB, MS, DG, GB), EarthCARE-FR, BAIVEC ESA-funded project (GB) and AOS-FR (MS, DG)
EU Horizon Europe REALISTIC, GA nº 101086690 (MS, DG)

### Author contributions

SK conceived the study and wrote the manuscript. MS and DG provided Maido lidar data. TL and FC provided TMF lidar data. TS, JY, RQ and BL provided Lauder lidar data, SK and SGB provided OHP lidar data. NB and VY processed AZOR data. GT provided OMPS-LP data. NK, AF and AH processed ATLID data.GB, TL, CS conducted balloon flights. All authors contributed to the final manuscript.

**Competing interests**: The authors declare that they have no competing interests.

### Data availability

EarthCARE ATLID data are available at https://ec-pdgs-dissemination1.eo.esa.int/oads/access/collection/EarthCAREL1Validated/tree; OMPS-LP data are available at https://snpp-omps.gesdisc.eosdis.nasa.gov/data/NOAA21_OMPS_Level2/OMPS_N21_LP_L2_AER_DAILY.1.0/; SAGE III data are available at https://asdc.larc.nasa.gov/project/SAGE%20III-ISS/g3bssp_6; TROPOMI data are available at https://dataspace.copernicus.eu/explore-data/data-collections/sentinel-data/sentinel-5p; GloSSAC data are available at https://asdc.larc.nasa.gov/project/GloSSAC/GloSSAC_2.22; NDACC lidar data are available at https://ndacc.larc.nasa.gov; AZOR data are available upon request.

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
