# Peer review of "Global transport of stratospheric aerosol produced by Ruang eruption from EarthCARE ATLID, limb-viewing satellites and ground-based lidar observations"

_EGUsphere, 2025_

## Author Response (AR2)

**Reply to Reviewer #1.**

**We thank the Reviewer #1 for the detailed review and critical remarks, which have all been carefully accounted for in the revised manuscript.**

*General issues:*

**Derivation of stratospheric AOD from Lidar observations**

The paper applied methodologies which obviously have been applied since a long time to investigate stratospheric aerosol distortions by volcanic aerosol. The scattering ratio of the ground-based and space borne lidars is used together with an assumed lidar ratio of 50 sr to estimate the stratospheric AOD. However, is this methodology still up to date?

First of all, the assumption of a lidar ratio of 50 sr at 355 nm is not discussed by means of any reference. Furthermore, it is in my opinion not justified per se as it assumes the existence of volcanic sulphate aerosol only and neglects any influence of wildfire smoke which was recently observed to be entered frequently in the stratosphere (see e.g. Khaykin et al., 2020; Ansmann 2024, Peterson 2025). Thus, justification for this lidar ratio and an uncertainty estimate for the derived products need to be done and are key for publication.

**In order to properly constrain the lidar ratio for sAOD calculation from ATLID L1 data, we have revised the analysis by using the daily/zonal average lidar ratios from ATLID L2 data (Fig. R1.1). The description of sAOD derivation in Sect. 2.1 has been revised accordingly. The updated time-latitude sAOD variation in Fig. 2b shows a better agreement with SAGE III extinction data, in particular regarding the wildfire-induced perturbation during the boreal Summer 2025, which is to be expected given the higher lidar ratio for stratospheric smoke aerosols.**

[Figure]

**Figure R1.1 Time-latitude variation of daily/zonal mean lidar ratio from ATLID L2 EBD product (BA baseline).**

Second and maybe more important, many of the lidars, especially the novel EarthCARE lidar ATLID which is used intensively, can provide extinction measurements directly. Why don't you use these products to derive the stratospheric AOD? While the ground based lidar may struggle with low-signal-to-noise ratio, EarthCARE at least is for sure capable to retrieve extinction profiles in the stratosphere. It is not clear to me why the authors use the detour via the scattering ratio and the assumed lidar ratio instead of using the directly measured extinction coefficient profiles. This is especially important also in view of the other data sources they use, as many provide extinctions measurements (e.g. NOAA-21 OMPS-LP, ISS SAGE III, GloSSAC......) and not scattering ratio.

**Indeed, the HSRL technique exploited by ATLID is capable of direct measurement of extinction. The reason why we opted for the L1B backscatter data is that the L2A EBD (BA) product exhibits important latitude and altitude dependent bias regarding the extinction, as illustrated in the figure R1.2. As this bias may be reduced or eliminated in the future ATLID L2A baselines, we refrain from presenting L2A-derived stratospheric aerosol extinction in this study. Instead, we use a combination of L1B backscatter and L2A lidar ratio to derive sAOD, which is shown to be in good agreement with reference-grade SAGE III extinction observations.**

[Figure]

**Figure R1.2 Comparison of ATLID L2 (BA baseline) stratospheric extinction (left) versus SAGE III V6.0 extinction (right) as a function of latitude and altitude for January through March 2025 period.**

Third, the computation of the scattering ratio from EarthCARE Level1b data is not understandable for me. As level 1b signal still suffer from molecular and particular attenuation (therefore attenuated backscatter coefficient), the simple integral of the sum of two signals multiplied with a lidar ratio seem to be not appropriate in my opinion. Please justify and describe more intensively.

**The computation of the scattering ratio ($R$) does not require any integral operations. It is computed as a sum of the attenuated Mie co-polar ($\beta_{M,\parallel}^{att}$), Mie cross-polar ($\beta_{M,\perp}^{att}$) and Rayleigh backscatter divided by the attenuated Rayleigh backscatter ($\beta_R^{att}$). Because the Rayleigh and Mie channels experience the same atmospheric attenuation, the two-way transmission term $T^2$ cancels exactly in the ratio, as follows from Eq. 1. Consequently, $R$ represents the true, unattenuated scattering ratio, independent of the extinction along the lidar path.**

**To derive the non-attenuated total Mie backscatter $\beta_M$ from $R$, we use the non-attenuated Rayleigh backscatter $\beta_R$ computed from the meteorological data provided with L2A product (Eq. 2). The SAOD is then computed as the total Mie backscatter vertically integrated between the zonally-averaged tropopause altitude $Z_{trop}$+1 km and**

**30 km and multiplied by the lidar ratio ($S$), derived from L2A product as the zonal-mean stratospheric layer average.**

**We have revised Sect. 2.1 entirely to better explain the derivation of non-attenuated scattering ratio and SAOD from the ATLID Level 1 attenuated Rayleigh and Mie measurement channels.**

**Some other methodological weaknesses are:**

The estimated mass of the emitted SO2 is from personal communication only without giving any reference. In my opinion, this is not sufficient as the value is key for Fig. 6 and the respective discussion. Thus, either you describe properly how the mass is estimated or give proper reference. Otherwise, you need to cancel the whole discussion about the impact of the recent eruptions compared to previous ones.

**The reference to the estimated mass of the emitted SO2 has been updated:**

Carn, S. (2025), Multi-Satellite Volcanic Sulfur Dioxide L4 Long-Term Global Database V4, Greenbelt, MD, USA, Goddard Earth Science Data and Information Services Center (GES DISC), Accessed: [19/11/2025], 10.5067/MEASURES/SO2/DATA405

The ground-based instruments and data retrieval descriptions are very heterogenous. Please unify and justify the different methodologies you use for each lidar system. Currently it seems that each ground-based lidar has its own retrieval algorithm with its own assumptions. This might be ok, but needs to be discussed. As for EarthCARE (described above) the same question comes up for many of the derived products from the ground based lidars: How do you account for attenuation of the backscatter when calculating the SAOD from scattering ratio.

**The description of ground-based lidars and data retrievals has been homogenised across the stations. The description of the ATLID-derived parameters has been revised. In particular, it has been pointed out the scattering ratio derived from ATLID L1 product is the same quantity as derived from ground-based lidar inversion.**

**Other general issues:**

The discussion often (not always) assumes Ruang aerosol to be the only reason for increased aerosol in the stratosphere neglecting other potential sources like wildfires, other volcano eruptions etc… this needs to be discussed better.

**While the discussion naturally focuses on the transport of Ruang aerosols provided the scope of the study, all the other factors of SAOD variability during ATLID era (late decay of Hunga, Antarctic and Arctic PSCs and the 2025 panboreal wildfire outbreak) are properly acknowledged.**

Related to this: In Fig. 6, you show major events for stratospheric AOD distortion, but I wonder if it is complete or is missing some of the severe Northern hemispheric fire events having a stratospheric impact (e.g. see in Ohneiser et al., 2023 for the Canadian fires in 2017 or many others as listed in Peterson et al 2025.). You state it by yourself in line 453: "Since 2017, the significant SAOD perturbations, caused by either volcanic eruptions or wildfire outbreaks in both hemispheres,

occurred at least once per year, maintaining the global stratospheric aerosol loading well above the background levels." But I can see only one fire event in the NH in Fig. 6.

**The GloSSAC-derived time series in Fig. 6 represent the area-weighted monthly-mean SAOD of the so-called stratospheric overworld (Holton et al., 1995), that is above the 380 K isentropic surface located above the tropopause at all latitudes. The residence time of aerosols in the stratospheric overworld is not limited by cross-tropopause stratosphere-troposphere exchange at midlatitudes and sub-tropics. The persistence of sulphate or carbonaceous aerosols above 380 K level is what translates into hemispheric-scale effects on climate, polar vortex chemistry and ozone layer.**

**As far as the stratospheric impact of increasingly intense wildfires is concerned, its magnitude and longevity at hemispheric scale largely depend on the efficiency of self-lofting mechanism, which is in turn driven by the composition of the plume and environmental meteorology. The variation of the overworld SAOD at hemispheric scale readily reflects the largest wildfire outbreaks that produced long-lived self-lofting anticyclones (SCV), specifically, the Canadian PNE and Australian ANYSO events.**

Please also broaden your view in the discussion and do not only cite yourself for explanation of certain events. There are many other scientists working on these topics and this would give the paper a broader justification.

**The referencing of the relevant literature has been broadened.**

Figs 4 and 5: If you would provide extinction profiles from the lidars, you could even compare these to the other data sources you use (e.g. NOAA-21 OMPS-LP, ISS SAGE III, GloSSAC......) which would value the paper even more.

**Figs. 4 and 5 show comparison between ATLID and collocated ground-based lidars measurements of scattering ratio. This quantity is derived directly from ATLID L1B product and represents the true non-attenuated scattering ratio, directly comparable to what is retrieved from elastic ground-based lidars using Klett-Fernald inversion.**
**As far, as the quantitative intercomparison of the extinction (using ATLID L2 products), this is outside the scope of this paper, as mentioned in Sect. 3.6.**

Many abbreviations are not explained, e.g., JPL, LA, JRA, PNE, AZOR ... It is good practise to write the whole name at the first instance.

**All the abbreviations have been developed except AZOR, which is a transliteration of a Russian abbreviation for this instrument.**

*Specific comments:*

- 100: calibrated backscatter profiles is not specific enough, please be more precise, the official nomenclature is "fully processed, calibrated, and geolocated attenuated backscatter signals" according
  to ttps://www.esa.int/Applications/Observing_the_Earth/FutureEO/EarthCARE/EarthCARE_data_products

**Corrected**

- 108: Again, it is not the Mie backscatter, but the attenuated Mie backscatter (coefficient)

**Corrected**

- 109: As written above, the use of a constant lidar ratio of 50 sr need to be justified and uncertainty estimates give unless it is obsolete because you use extinction profile observations.

**We have switched to the ATLID-derived lidar ratio as explained above.**

- 110: The formula used is not clear to me. How do you account for example for the attenuation of the molecular extinction? In Level 1B data there is not a particle backscatter coefficient which but the attenuated one. Can you justify your formula? But maybe I am also wrong.

**The formula has been revised to explain why the L1-derived scattering ratio is non-attenuated.**

- 126: If you assume a lidar ratio of 50 sr for Maido as well, at least you cannot validate your assumptions - as you make the same ones - and it is not clear if the retrieved SAOD is valid. But as far as I see you do not use the AOD from the ground-based observations generally?

**The comparison with ground-based lidars is provided for SR only, not for SAOD.**

- 142: "The backscatter coefficient is derived from the backscatter ratio using the 3-hourly atmospheric density output from MERRA-2 interpolated at the site." it is not clear for me how this works and which assumptions are made. Please describe more explicitly.

  **The description of ground-based lidars and data retrievals has been homogenised and improved.**

- 162: Why using lidar ratios of 33 to 58? Is this the natural variability? What does this mean for your SAOD retrievals?

**The choice of LR is discussed by Sakai et al. (2025). The comparison with ground-based lidars is provided for SR only, not for SAOD.**

- 167: How can you justify the backscatter Angstroem exponent of 1.8 ?

**The BAE value of 1.8 is justified by referring to the paper by Chouza et al. (2020) and references therein.**

- 192: please describe more explicitly what the application of a "5 point boxcar filter" means. I.e., what final effective vertical and horizontal and temporal resolution you have.

  **The 5-point boxcar smoothing (or moving average) applied to a profile on a 100 m vertical grid would result in an effective vertical resolution of 500 m. It does not have any effect on the horizontal or temporal resolutions.**

- 202: Please state which wavelength you use for the comparison.

**Done.**

- 239 ff. Where does this information come from? Any reference?

**The reference is provided: Global Volcanism Program (2025).**

- 244: What is the extinction ratio? It's not explained in the methodology. Why not using extinction coefficient profiles?

**Extinction ratio (ER, ratio of aerosol to molecular extinction) is a commonly used parameter in stratospheric studies. It is proportional to the aerosol mixing ratio and is thus convenient for detection of meridional transport in the stratosphere.**

- 250: "DU" never explained as many other abbreviations – I assume DU is not a SI unit and thus need to be explained.

**While DU (Dobson unit) is not an SI unit, we assumed that a reader of the ACP journal would not be wondering what it means when applied to the $SO_2$ total column.**

- 251: The estimate of the mass of the SO2 plume is given without justification or reference, this is inappropriate given its importance.

**The reference to Carn, 2025 has been provided.**

- 264: Fig. 1 Date for 1d is wrongly stated in the caption.

**Corrected.**

- 276: You state that ATLID might have captured PSC, but then the assumed Lidar ratio is not valid anymore, right?

**We have switched to the ATLID-derived lidar ratio for SAOD analysis as explained above.**

- 283ff: The SAOD perturbation could be also partly affected by stratospheric wildfire smoke - please discuss this. Such events not only occurred in 2025 as exemplarily stated in line 291 ff. Furthermore, can you exclude any other major source of stratospheric aerosol? I guess yes, but you need to discuss this.

**The SAOD perturbation of the PWO-2025 event in May 2025 is discussed in this section. Other than that, we do not expect any measurable stratospheric impact of wildfires during Boreal winter in the Northern hemisphere. All the events that have had a measurable stratospheric impact are listed on GSAW web portal. The events in 2024 are listed in BAMS State of the Climate 2024 (Chapter 2.5).**

- 306: Again, when using ATLID and linking the observations to PSC the used Lidar ratio might not be appropriate. At least, error estimates should be made.

We have switched to the ATLID-derived lidar ratio for SAOD analysis as explained above.

- 310: "strong PSC signal": Can you exclude other sources for this strong signal? Please discuss.

A strong enhancement of aerosol abundance in the Winter Polar stratosphere can only be explained by the PSC. This is well a well-documented phenomenon (see e.g. the review papers by Tritscher et al., (2021) and Kremser et al., (2016) (both cited in the paper) and we believe it is unnecessary to discuss this in a paper focusing on stratospheric aerosols.

- 319: "However, in the northern extratropics ATLID shows reduced sensitivity to the volcanic aerosol layer, for which the reason remains to be investigated" – this statement is just an assumption. I would rephrase it to e.g.:"… ATLID shows lower aerosol load compared to OMPS…".
  Concluding on a reduced sensitivity is in my opinion not valid from the figures you show.

We rephrased the sentence according to the suggestion.

- 355: "The upper part of ATLID profiles, above ~25 km tends to be noisier and prone to noticeable deviations with respect to the ground-based data within 1-2 km-thick layers": this is too strong statement. Maybe the ground-based reference is also not seeing some of the aerosol peaks? I recommend to formulate it more neutral. At least I cannot find evidence for the statement in the figure.

We do not see any reasons to assume that the ground-based lidar measurements with higher vertical resolution would miss local maxima reported by ATLID. As the SNR naturally decreases with height, it is conceivable that ATLID's 10 seconds-long acquisition could exhibit increased noise. This is particularly evident from ATLID data above 30 km (not shown). Nevertheless, we have rephrased the sentence to make it more neutral: *"The upper part of ATLID profiles, above ~25 km exhibits some discrepancies relative to the ground-based data within 1–2 km-thick layers."*

- 385: "remnants of aerosols produced by the Hunga eruption in January 2022," à or other aerosols like smoke….

It is very unlikely that the subtle enhancement at 26 km detected at TMF in September 2024 could be associated with smoke. The only known wildfire outbreak that resulted in self-lofting of smoke aerosols up to this altitude is the Australian "Black Summer" (ANYSO) event in 2019/2020, however its impact was limited to the Southern hemisphere, whereas by September 2024, these aerosols have long been removed from the atmosphere.

- 386: Why can you undoubtedly link this to Ruang aerosol?

Because there are no other sources of stratospheric aerosols at this time, altitude and latitude, as follows from the analysis of satellite observations. We removed the word "undoubtedly" to smoothen the statement

- 394-395: Why can the lower aerosol layer be linked to Ruang and the upper one to Hunga volcano? Please explain in more detail.

**This sentence has been removed as we are not entirely sure in this interpretation.**

- 398: What do you mean with regional feature? Can you describe in more detail?

**The difference between the local and zonal mean observations suggests that the feature could only be present at the regional scale.**

- Sec 3.4 The balloon launched are temporally and geographically partly very close to OHP observations. Maybe you can use this to discuss? E.g., Figure 5 h and i.

**The distance between the balloon launch location in Orleans and OHP is around 500 km, which is much larger than the collocation criteria used here for ATLID validation.**

- 412: "upper enhancement around 26 km may be the result of the poleward transport of the volcanic aerosols from the tropics" what other volcanic aerosols? Please discuss.

**This statement has been removed as per other reviewer's comment.**

- 460: Based on this statement (factor 1.8 increase), can you exclude any other event contributing (other volcanos, wildfires)? Please discuss.

**An event causing stratospheric aerosol perturbation that would be noticeable in the hemispheric-scale SAOD series would certainly be noticeable in the time-latitude section of SAOD. So yes, we can exclude with certainty any other contribution to the Ruang-driven factor of 1.8 increase in SAOD.**
**All the events that have had a measurable stratospheric impact are listed on GSAW web portal. The events in 2024 are listed in BAMS State of the Climate 2024**

- 464: The introduction of ACE comes a bit out of nothing. Here some more explanation is needed about this experiment and what was done.

**The text regarding ACE observations of the young plume has been moved backward.**

- 475: Ansmann et al 2022, Ohneiser et al 2022, and Solomon et al. 2023, discussed the influence of smoke on the ozone hole. You should mention this as well in view of your future investigations. As marked in the general statements, the disentangling of smoke and volcanic aerosol is still not clear and need to be discussed as well

**Suggested citations have been added.**

- 480: "In Spring…": one has the feeling the discussion is still wrt northern hemisphere. If so, the reference to the figures is not appropriate. If this is not the case, the sentence should be rephrased.

**The reference to Fig. 4 has been removed as the discussion regards indeed the Northern hemisphere and Boreal Spring.**

**Reply to Reviewer #2.**

**We thank the Reviewer #2 for the positive evaluation and useful remarks.**

This is a nice paper that should be published after a few modifications. The one difficult claim, which is not well supported by the discussion or the figures, is that there was transport from the tropics into the Northern Hemisphere along the 750 K isentrope. Here are the detailed comments.

313-315 This claim by the authors is not substantiated by either Figs 3 e) or f). The northward extension of scattering ratios near 1 (purple) are not at all obvious and certainly not along the 750 K contour.

412-414 This seems like a stretch given there was no clear evidence of transport from the tropics along the 750 K isentrope. Could this be a filament from the polar vortex as it dissipates in the spring? Where did the air come from? Were there any back trajectory calculations?

**Indeed, the evidence provided may be insufficient for such inference. We opted to remove it.**

Other comments/corrections.

Figure 1a) b) Where is the triangle marking Ruang?

**The triangle visibility has been improved.**

276 … Antarctic …

**OK**

280-282 This statement is perhaps reasonable at SAOD > 15e-3, but less than this the difference is a factor of two or more, very obvious in the NH mid latitudes. Might be worth mentioning the regions of particularly larger discrepancies.

**The SAOD retrieval from ATLID L1B data has been revised and updated with the latest baseline, which reduced the differences, in particular at NH midlatitudes.**

281-283 This claim is questionable. Yes there is a faint signal in OMPS-LP, but this is not really supported by either SAGE III or ATLID.

**The northern poleward propagation of aerosols is not obvious from the time-latitude SAOD pattern, in particular because of the highly-variable tropopause altitude in the subtropical "surf" zone, which creates an apparent discontinuity in stratospheric aerosol abundance between the tropics and**

**midlatitudes in both hemispheres. Otherwise, the presence of Ruang aerosols in the northern extratropics is cross-confirmed by ATLID and NDACC lidars.**

Figure 2 Why does the SAGE III data stop in July. Just not available yet?

**All the data sets, including SAGE III have been updated.**

319 Probably the reason is because it wasn't there.

**Distinct local maxima of scattering ratio exhibited by the collocated profiles in NH provide a strong support for the presence of Ruang aerosols at these latitudes, which is corroborated by OMPS-LP and SAGE III observations.**

338 Isn't it a black dotted line?

**This regards Fig. 4 not Fig. 3.**

341 The structure of the zonal mean is not vastly different, it just doesn't capture the fine scale structure, nor would a zonal mean be so expected.

**Sentence rephrased.**

351 Isn't it the opposite with the zonal mean exhibiting somewhat lower scattering ratios than the local measurements?

**Yes, thank you, correction made.**

455 Is there a reference for 0.4 Tg of sulfur?

**Yes, the proper reference has been included.**